# ViSpec: Accelerating Vision-Language Models with Vision-Aware Speculative Decoding

**Jialiang Kang**[1,2][*]  **Han Shu**[2]  **Wenshuo Li**[2]  **Yingjie Zhai**[2]  **Xinghao Chen**[2][†]

[1]Peking University

[2]Huawei Noah's Ark Lab

jkang@stu.pku.edu.cn

{han.shu,liwenshuo,zhaiyingjie,xinghao.chen}@huawei.com

## Abstract

Speculative decoding is a widely adopted technique for accelerating inference in large language models (LLMs), yet its application to vision-language models (VLMs) remains underexplored, with existing methods achieving only modest speedups ($< 1.5\times$). This gap is increasingly significant as multimodal capabilities become central to large-scale models. We hypothesize that large VLMs can effectively filter redundant image information layer by layer without compromising textual comprehension, whereas smaller draft models struggle to do so. To address this, we introduce **Vision-Aware Speculative Decoding (ViSpec)**, a novel framework tailored for VLMs. ViSpec employs a lightweight vision adaptor module to compress image tokens into a compact representation, which is seamlessly integrated into the draft model's attention mechanism while preserving original image positional information. Additionally, we extract a global feature vector for each input image and augment all subsequent text tokens with this feature to enhance multimodal coherence. To overcome the scarcity of multimodal datasets with long assistant responses, we curate a specialized training dataset by repurposing existing datasets and generating extended outputs using the target VLM with modified prompts. Our training strategy mitigates the risk of the draft model exploiting direct access to the target model's hidden states, which could otherwise lead to shortcut learning when training solely on target model outputs. Extensive experiments validate ViSpec, achieving, to our knowledge, the first substantial speedup in VLM speculative decoding. Code is available at https://github.com/KangJialiang/ViSpec.

## 1   Introduction

The success of large language models (LLMs) has spurred the development of vision-language models (VLMs) capable of processing and generating content from both visual and textual inputs. Recent VLMs, such as LLaVA-NeXT (LLaVA-1.6) [15] and Qwen2.5-VL [1], demonstrate impressive performance in tasks including image captioning, visual question answering, and multimodal dialogue. However, as VLMs increase in scale and complexity, their inference times grow substantially, posing significant challenges for practical deployment.

Speculative decoding [13] has proven effective in accelerating LLM inference by employing a smaller, faster draft model to propose candidate token sequences, which the larger target model verifies in parallel. Correct predictions from the draft model enable the target model to skip costly autoregressive computations, resulting in significant speedups. While speculative decoding is well-established for

---

[*]Work done during an internship at Huawei Noah's Ark Lab.

[†]Corresponding author.

39th Conference on Neural Information Processing Systems (NeurIPS 2025).

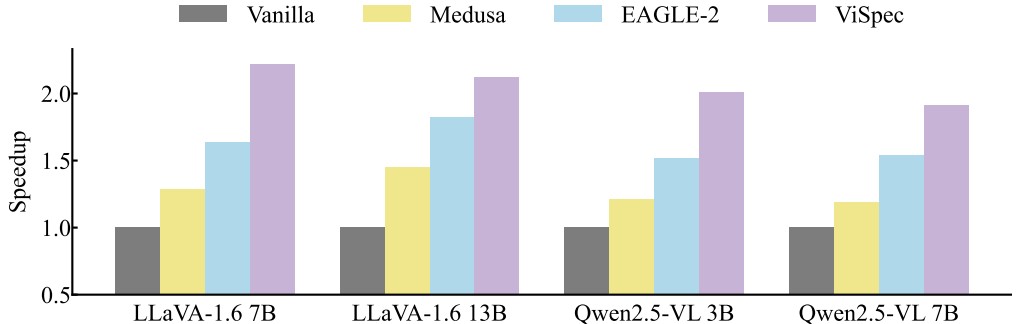

Figure 1: Speedup ratios of various methods at temperature = 0, evaluated on the GQA test set using four VLMs: LLaVA-v1.6-Vicuna-7B, LLaVA-v1.6-Vicuna-13B, Qwen2.5-VL-3B-Instruct, and Qwen2.5-VL-7B-Instruct.

LLMs, its application to VLMs remains underexplored, with prior approaches [8, 12] achieving only marginal speedups. We attribute this limitation to fundamental differences between textual and visual data. Text, honed over centuries, is abstract and information-dense, whereas images, despite their visual richness, often contain considerable redundancy. Consequently, small draft models struggle to extract pertinent visual information while preserving textual coherence in VLMs.

To address this challenge, we propose **Vision-Aware Speculative Decoding (ViSpec)**, a novel speculative decoding framework designed specifically for VLMs. ViSpec incorporates a lightweight vision adaptor module to compress numerous image tokens into a compact, informative representation. These compressed tokens are seamlessly integrated into the draft model's attention layers, retaining the original image's positional information. Furthermore, drawing inspiration from target-aware feature injection in EAGLE [19, 18, 20], we extract a global feature vector for each input image and augment all subsequent text tokens with this feature until the next image is encountered. This mechanism equips the draft model with robust global visual context, enhancing prediction accuracy.

A significant obstacle in developing speculative decoding for VLMs is the scarcity of large-scale, publicly available multimodal datasets with extended assistant responses. To overcome this, we repurpose existing datasets by modifying prompts and leveraging the target VLM to generate long responses, thereby creating synthetic training data. Although the draft model could potentially exploit access to the target model's hidden states during training, the randomness in the target model's sampling strategy and our adoption of multi-token prediction, inspired by DeepSeek [21], effectively mitigate this risk.

Our experiments demonstrate that ViSpec significantly outperforms existing speculative decoding methods for VLMs, achieving substantial speedups without compromising generation quality. To our knowledge, this work represents the first meaningful acceleration of VLM inference through speculative decoding.

Our main contributions are as follows:

- We introduce Vision-Aware Speculative Decoding (ViSpec), a speculative decoding framework tailored for VLMs.

- We propose dual integration mechanisms—attention integration and feature augmentation—to enable a small draft model to efficiently incorporate visual context.

- We develop a training strategy that extends existing vision-language datasets to include long-response tasks, leveraging multi-token prediction.

- We empirically validate ViSpec on four popular VLMs, achieving notable speed improvements and establishing the first practical acceleration in this domain.

## 2 Related Work

### 2.1 Speculative Decoding

Speculative decoding [13] accelerates inference in LLMs by utilizing a smaller draft model to propose candidate token sequences, which the target model verifies in parallel. This method achieves speedups of 3–4× on text-only tasks while preserving output fidelity [13]. Subsequent advancements have refined this approach. Self-speculative decoding [36] derives the draft model from the target model, minimizing training overhead through shared parameters. On-the-fly adaptation methods, such as SwiftDecode [33], dynamically adjust the draft model during inference to adapt to varying input distributions, enhancing robustness across tasks. Specifier [25] employs an ensemble of small models for parallel draft generation, increasing prediction diversity and speedup. Cascade Speculative Drafting [5] uses a sequence of draft models with increasing complexity to balance speed and accuracy, achieving up to 3.5× speedup on large-scale LLMs. Medusa [2] integrates multiple decoding heads into the target model's architecture, eliminating the need for a separate draft model while maintaining comparable performance. The EAGLE series [19, 18, 20] improves draft predictions by injecting target-aware hidden states, aligning the draft model closely with the target model's output distribution. A concurrent work, EAGLE-3 [20], adopts a multi-token prediction strategy termed *training-time test*, though its performance gains depend heavily on scaling training data, with limited improvements when dataset size is fixed. Additionally, SpecTr [29] optimizes token acceptance rates using optimal transport, while REST [10] enhances draft predictions with retrieval-based external knowledge, excelling in knowledge-intensive tasks. Recent surveys [34] offer detailed analyses of speculative decoding techniques, discussing their trade-offs and open challenges.

### 2.2 Vision-Language Models

Vision-language models (VLMs) integrate visual and textual inputs to address tasks such as image captioning, visual question answering, and multimodal dialogue. Recent advancements have led to state-of-the-art models like LLaVA-NeXT [15] and Qwen2.5-VL [1], which combine powerful vision encoders, such as CLIP [26], with large-scale LLMs to achieve superior performance across diverse applications. Models like BLIP-2 [16] and MiniGPT-4 [38] leverage pretrained vision and language components with learnable interfaces to bridge modality gaps, enabling efficient multimodal processing. However, the computational complexity of processing high-dimensional image inputs, coupled with the autoregressive nature of text generation, results in significant inference latency, posing challenges for real-time deployment. Efforts to address these issues include efficient vision encoders, such as those proposed in EVA-CLIP [28], and optimized training strategies that reduce memory overhead. Despite these advances, inference efficiency remains a critical bottleneck, motivating the exploration of speculative decoding for VLMs.

### 2.3 Speculative Decoding for Vision-Language Models

The application of speculative decoding to VLMs is an emerging area with limited prior work. The only notable effort, by [8], applied speculative decoding to LLaVA-7B using a small language-only draft model, achieving up to 1.5× speedup. Their experiments with a small VLM draft model incorporating an image encoder yielded only marginal gains, highlighting the challenge of effectively processing visual information in the draft model due to the high redundancy and computational complexity of image inputs. These limitations highlight the need for specialized frameworks that can effectively integrate visual and textual information in draft models while maintaining high prediction accuracy. Our proposed ViSpec framework addresses these challenges by introducing vision-aware mechanisms to enhance the draft model's ability to process multimodal inputs efficiently.

## 3 Preliminaries

### 3.1 Speculative Decoding

Speculative decoding [3, 13, 25, 29] is a lossless acceleration technique for LLMs that alternates between a drafting stage and a verification stage to expedite autoregressive decoding. Let $t_i$ denote the $i$-th token in a sequence, and let $T_{a:b} = \{t_a, t_{a+1}, \ldots, t_b\}$ represent a token sequence. Given a prefix $T_{1:j}$, speculative decoding proceeds as follows: in the drafting stage, a lightweight draft

model autoregressively generates a sequence of $k$ tokens, $\hat{T}_{j+1:j+k}$, along with their probabilities $\hat{p}_{j+i}(\hat{t}_{j+i})$ for each token $\hat{t}_{j+i}$. In the verification stage, the target model evaluates $\hat{T}_{j+1:j+k}$, computing its own probabilities $p_{j+i}(\hat{t}_{j+i})$. Each draft token $\hat{t}_{j+i}$ is accepted with probability $\min\left(1, \frac{p_{j+i}(\hat{t}_{j+i})}{\hat{p}_{j+i}(\hat{t}_{j+i})}\right)$. If a token is rejected, a new token is sampled from the normalized distribution $\text{norm}\left(\max\left(0, p_{j+i}(\hat{t}_{j+i}) - \hat{p}_{j+i}(\hat{t}_{j+i})\right)\right)$, and subsequent draft tokens are discarded. This process ensures that the generated sequence maintains the same probability distribution as that produced by the target model without acceleration, preserving distributional consistency. We enhance this approach by adopting the context-aware dynamic draft tree from EAGLE-2 [18], an improvement over the draft tree in [25], which enables the draft model to generate multiple candidate tokens per position, facilitating more efficient exploration of the token space.

## 3.2 Vision-Language Models

Modern VLMs [1, 15, 16] typically extend a base large language model (LLM) by incorporating visual information through a vision encoder. Formally, given an input image $I$, a vision encoder $\mathcal{E}_v$ maps it to a sequence of visual embeddings $V_{1:r} = \mathcal{E}_v(I) \in \mathbb{R}^{r \times d}$, where $r$ denotes the number of visual embeddings and $d$ is the embedding dimension. Let $\mathcal{E}_t$ represent the text embedding layer of the LLM. For a multimodal input sequence comprising both visual and textual tokens, the joint input representation is constructed as $H_{1:n} = \mathcal{E}_t(T_{1:k}) \oplus V_{1:r} \oplus \mathcal{E}_t(T_{k+1:j})$, where $\oplus$ denotes sequence concatenation. The VLM processes this hybrid sequence autoregressively. Notably, the LLM architecture remains unchanged; the only modification is the inclusion of visual embeddings $V_{1:r}$ within the input sequence. Since the output space remains the text token vocabulary $\mathcal{V}$ and the autoregressive generation mechanism is preserved, speculative decoding methods designed for LLMs can, in principle, be directly applied to VLMs by treating visual embeddings as part of the input context. Formally, for any prefix containing visual embeddings $V_{1:r}$ and text tokens $T_{1:j}$, the speculative decoding procedure outlined in Sec. 3.1 remains valid, with probabilities $p_{j+i}$ and $\hat{p}_{j+i}$ implicitly conditioned on $V_{1:r}$.

## 4 Method

### 4.1 Overcoming Redundancy: Image Embedding Compression

In speculative decoding, the draft model is typically a smaller, shallower version of the target model. We demonstrate that a single-layer Transformer-based draft model is fundamentally limited in processing long, redundant sequences, particularly when redundant image patches (e.g., uniform color blocks) dominate the input.

Consider a sequence of $R + 1$ image and text embeddings $e_1, \ldots, e_{R+1} \in \mathbb{R}^d$, where $R$ embeddings are identical, i.e., $e_{r_1} = \ldots = e_{r_R} = s$, and a unique token at position $u$ has embedding $e_u = t$. The Transformer has a single self-attention layer with weight matrices $W_q, W_k, W_v \in \mathbb{R}^{d \times d}$. Ignoring positional encoding and attention scaling for simplicity, the output at position $i$ is:

$$y_i = \sum_{j=1}^{R+1} \alpha_{ij} v_j, \quad \text{where} \quad \alpha_{ij} = \frac{\exp\left(q_i k_j^\top\right)}{\sum_{k=1}^{R+1} \exp\left(q_i k_k^\top\right)}, \tag{1}$$

with $q_i = W_q e_i$, $k_j = W_k e_j$, and $v_j = W_v e_j$. For the $R$ redundant tokens, we have:

$$q_i k_r^\top = (W_q e_i)(W_k e_r)^\top = W_q e_i s^\top W_k^\top, \tag{2}$$

which is identical across all redundant tokens. As $R$ increases, the attention weight to the unique token becomes:

$$\alpha_{iu} = \frac{\exp\left(B\right)}{R \exp\left(A\right) + \exp\left(B\right)}, \tag{3}$$

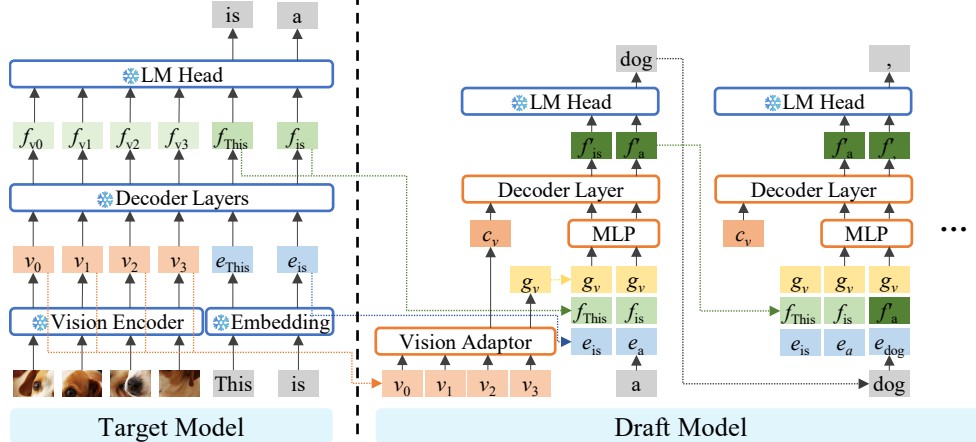

Figure 2: Overview of the ViSpec framework. Given an input image and text prompt, ViSpec compresses image tokens using a lightweight vision adaptor to produce a small set of visual tokens. These tokens are prepended to the text input and fed into the draft model's attention mechanism. A global visual feature vector, extracted from the compressed image tokens, is injected into the draft model's text generation process. The figure illustrates two decoding steps of the draft model, where $f$ denotes the target model's last-layer hidden state, $f'$ the draft model's last-layer hidden state, $v$ visual embeddings, $e$ text embeddings, $c$ compressed image tokens, and $g$ the global visual feature vector.

where $A = W_q e_i s^\top W_k^\top$ is the score for redundant tokens, and $B = W_q e_i t^\top W_k^\top$ is the score for the unique token. As $R \to \infty$, the denominator is dominated by $R \exp(A)$, causing $\alpha_{iu} \to 0$. Meanwhile, $\alpha_{ir} \to \frac{1}{R}$ for each redundant token, so the output approximates:

$$y_i \approx \sum_{m=1}^{R} \frac{v_{r_m}}{R} = W_v s, \tag{4}$$

effectively averaging over the redundant tokens and neglecting the unique token. Furthermore, it has been proven theoretically that a $K + 1$ layer network is required to handle a nesting complexity of $K$ [30], indicating that shallow draft models struggle to extract useful information from long, redundant image embeddings, thus constraining their effectiveness in speculative decoding for VLMs.

The limitations of shallow draft models in processing multimodal sequences necessitate a specialized approach for speculative decoding in VLMs. Drawing on insights from [22], which emphasize the critical role of draft token generation speed in achieving end-to-end speedup, we propose a lightweight Q-Former-inspired [16] vision adaptor (see Fig. 3). This module utilizes a lightweight Transformer encoder with a fixed set of learnable query vectors. The visual features extracted from the input image serve as key and value inputs to the Transformer's attention layers, while the learnable query vectors function as queries. Through this attention mechanism, each query vector selectively attends to relevant portions of the visual features, condensing them into a small set of compact feature vectors. These vectors, significantly fewer than the original embeddings, act as compressed visual embeddings. They are seamlessly integrated into the draft model's attention mechanism, preserving the positional information of the original image by maintaining relative spatial locations. By splitting the input into a concise image sequence and a compressed-image-plus-text sequence, we improve the draft model's efficiency in handling long multimodal sequences.

### 4.2 Addressing Lost-in-the-Middle: Global Visual Feature Integration

While image embedding compression can condense visual tokens into a compact sequence amidst a series of text tokens, this approach poses challenges in speculative decoding, which prioritizes long assistant responses. The *lost in the middle* effect [23], particularly pronounced in shallow models such as our draft model, causes performance degradation when critical visual information is situated in the middle of long contexts, leading to a U-shaped performance curve.

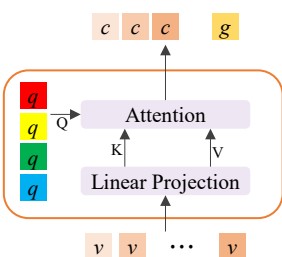

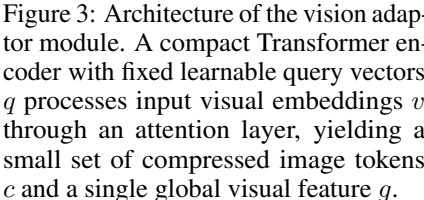

Figure 3: Architecture of the vision adaptor module. A compact Transformer encoder with fixed learnable query vectors $q$ processes input visual embeddings $v$ through an attention layer, yielding a small set of compressed image tokens $c$ and a single global visual feature $g$.

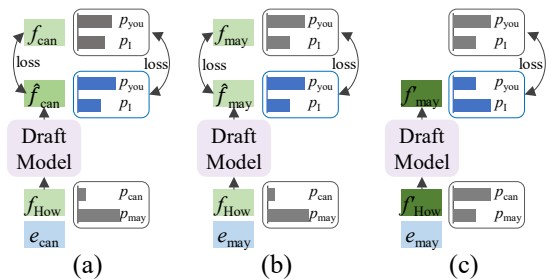

Figure 4: Comparison of training procedures: (a) EAGLE training, (b) training with greedy target model responses without multi-token prediction, and (c) ViSpec training. Here, $e$ denotes input embeddings, $f$ represents target model hidden states, $\hat{f}$ indicates EAGLE draft model hidden states, $f'$ denotes ViSpec draft model hidden states, and $p$ signifies token probabilities.

Although compressed visual embeddings provide a compact representation of images for the draft model, they may not fully capture the holistic visual context. As discussed in Sec. 4.1, simply increasing the number of image tokens is suboptimal, as shallow draft models lack the capacity to effectively attend to lengthy, redundant sequences. Moreover, as generated text sequences lengthen, image tokens become increasingly obscured within the text, exacerbating the *lost in the middle* effect [23] and undermining the draft model's ability to maintain consistent visual grounding. This often results in reduced coherence between the generated text and the input image. To address these challenges, we propose extracting a global feature vector from the input image and integrating it into each subsequent text token, ensuring persistent access to global visual context throughout the text generation process.

We derive the global feature vector from the final output of the vision adaptor module. This vector is transformed and incorporated into the hidden states of all subsequent text tokens in the draft model. Formally, at each text position $t$, we compute the augmented hidden state $f_t^{\text{aug}}$ as:

$$f_t^{\text{aug}} = f_t + W_g g, \tag{5}$$

where $f_t$ is the original hidden state, $g$ denotes the global visual feature vector, and $W_g$ is a learned projection matrix. This architectural enhancement equips the draft model with continuous visual context, enhancing its ability to generate accurate speculative tokens that maintain strong alignment with the input image across extended generation sequences.

### 4.3 Dataset Generation and Training

Training an effective draft model for speculative decoding requires a large, diverse dataset of high-quality target model outputs. For VLMs, this necessitates multimodal datasets with extended assistant responses, which are scarce in the public domain. To address this, we propose a novel data generation strategy that repurposes existing multimodal datasets, even those lacking long responses. We modify prompts in datasets such as visual question answering or image captioning to elicit longer, more descriptive responses from the target VLM. For instance, in visual question answering, we rephrase simple questions to request detailed explanations or reasoning. Similarly, for image captioning, we prompt the VLM to produce elaborate descriptions. This approach yields a robust synthetic training dataset without requiring manual annotation.

A potential concern is that the draft model might overfit to the target model's outputs by exploiting its hidden states during training. However, the randomness in the target model's sampling strategy, driven by a temperature parameter, and the adoption of multi-token prediction, as proposed by DeepSeek [21], mitigate this risk. We illustrate this in Fig. 4. In (a) EAGLE's text-only training, the target model is likely to generate "may" instead of "can" when the ground truth is "can." Here, the target model's hidden state $f_{\text{How}}$ acts as noisy augmentation, enabling the draft model to learn corrective behavior, as $f_{\text{can}}$ is conditioned on both $f_{\text{How}}$ and the embedding $e_{\text{can}}$. In (b) training with

greedy target model responses without multi-token prediction, the draft model's inputs (e.g., $f_{\text{How}}$, $e_{\text{may}}$) exhibit a one-to-one correspondence, and the supervision $f_{\text{may}}$ is conditioned solely on $f_{\text{How}}$ (ignoring previous inputs, as $f$ already contains most of the information), effectively reducing to a single-step Medusa [2]. In (c) ViSpec's training procedure, we avoid this issue by using sampling to disrupt one-to-one correspondences between hidden states $f$ and embeddings $e$. Additionally, we incorporate the draft model's own hidden states $f'$ as input, which serve a similar corrective role as in (a) when the target model deviates from the ground truth, allowing the draft model to learn self-correction without manually crafted datasets. We optimize the loss:

$$L = \text{CrossEntropy}(p_i, \hat{p}_i), \tag{6}$$

where $p_i$ and $\hat{p}_i$ denote the target model's and draft model's probabilities for the $i$-th token, respectively.

## 5 Experiments

### 5.1 Experimental Setup

**Hardware.** All experiments are conducted on a single GPU. Draft models are trained using 8x GPUs.

**Models.** We evaluate our proposed **Vision-Aware Speculative Decoding (ViSpec)** framework on four open-source vision-language models: LLaVA-v1.6-Vicuna-7B [15], LLaVA-v1.6-Vicuna-13B [15], Qwen2.5-VL-3B-Instruct [1], and Qwen2.5-VL-7B-Instruct [1]. We use their official weights and configurations from the Hugging Face Transformers library [32].

**Baselines.** We compare ViSpec against two established speculative decoding frameworks originally designed for language models: Medusa [2] and EAGLE-2 [18]. To adapt these frameworks for VLMs, we modify their input pipelines to process image patch embeddings from the VLM's original vision encoder, enabling the draft models to generate speculative tokens conditioned on both visual and textual contexts. This adaptation is feasible as both Medusa and EAGLE-2 rely on general token prediction mechanisms that are theoretically compatible with multimodal sequences, provided the draft model can handle visual inputs.

**Training Datasets.** We train the draft models for both baselines and ViSpec using a two-stage process. Initially, all draft models are trained on the ShareGPT dataset, comprising 68,000 dialogue iterations, to establish a robust text-based foundation. For multimodal training, we fine-tune the baseline draft models (Medusa and EAGLE-2) on 68,000 samples randomly selected from the LLaVA Visual Instruct Pretrain LCS dataset [15], enabling them to process visual inputs. For ViSpec, we augment this dataset with synthetic long assistant responses generated using the target VLM, as described in Sec. 4.3.

**Tasks.** We evaluate performance on eight diverse multimodal benchmarks: ScienceQA (SQA) [24], MM-Vet [35], MME [7], TextVQA [27], COCO Captions (COCO Caps) [4], VizWiz [9], GQA [11], and SEED-Bench [14]. These datasets cover tasks such as visual question answering, image captioning, and multimodal evaluation. To ensure generalizability, we use consistent model weights across all tasks without task-specific fine-tuning. Following [8], we design prompts to elicit long, detailed responses from the models.

**Metrics.** As ViSpec employs strict speculative decoding to preserve the target model's generation quality, quality evaluation is unnecessary. We focus on acceleration performance, measured using the following metrics:

- **Average Acceptance Length $\tau$:** The average number of tokens accepted from the draft model per drafting-verification cycle.
- **Speedup Ratio:** The ratio of inference time for standard autoregressive decoding to that for different speculative decoding methods.

### 5.2 Comparison with Baselines

Table 1 and Figure 1 present a comprehensive evaluation of ViSpec's acceleration performance compared to Medusa [2] and EAGLE-2 [18] across multiple vision-language models and tasks. The

Table 1: Speedup ratios and average acceptance lengths $\tau$ for different methods. For a fair comparison, we do not relax the draft token acceptance condition of Medusa under non-greedy settings as proposed in the original paper; instead, we adopt the same acceptance condition as EAGLE-2. Speedup ratios are computed based on the average time required to generate each token.

| Model | Method | SQA $\tau$ | Speedup | MM-Vet $\tau$ | Speedup | TextVQA $\tau$ | Speedup | MME $\tau$ | Speedup | COCO Caps $\tau$ | Speedup | VizWiz $\tau$ | Speedup | GQA $\tau$ | Speedup | SEED-Bench $\tau$ | Speedup | Avg. $\tau$ | Speedup |
|---|---|---|---|---|---|---|---|---|---|---|---|---|---|---|---|---|---|---|---|
| | | | | | | | | | Temperature=0 | | | | | | | | | | |
| LLaVA-1.6 7B | Medusa | 0.72 | 1.41x | 0.73 | 1.42x | 0.77 | 1.46x | 0.70 | 1.41x | 0.66 | 1.61x | 0.76 | 1.38x | 0.73 | 1.29x | 0.72 | 1.38x | 0.72 | 1.42x |
| | EAGLE-2 | 2.48 | 2.14x | 0.63 | 1.48x | 0.63 | 1.25x | 1.25 | 1.68x | 1.24 | 1.80x | 1.15 | 1.40x | 1.74 | 1.64x | 1.40 | 1.59x | 1.31 | 1.62x |
| | ViSpec | **2.86** | **2.37x** | **2.83** | **2.52x** | **2.95** | **2.90x** | **2.84** | **2.55x** | **3.30** | **3.22x** | **3.16** | **2.67x** | **2.88** | **2.22x** | **3.03** | **2.22x** | **2.98** | **2.58x** |
| LLaVA-1.6 13B | Medusa | 0.84 | 1.61x | 0.80 | 1.47x | 0.89 | 1.51x | 0.79 | 1.47x | 0.75 | 1.48x | 0.81 | 1.45x | 0.85 | 1.45x | 0.82 | 1.40x | 0.82 | 1.48x |
| | EAGLE-2 | 2.02 | 2.12x | 1.64 | 1.59x | 1.71 | 1.91x | 1.81 | 1.85x | 1.83 | 2.01x | 1.98 | 1.90x | 2.10 | 1.82x | 2.03 | 1.66x | 1.89 | 1.86x |
| | ViSpec | **2.76** | **2.57x** | **2.73** | **2.34x** | **2.78** | **2.43x** | **2.78** | **2.36x** | **3.18** | **2.82x** | **2.93** | **2.26x** | **2.95** | **2.12x** | **3.04** | **2.16x** | **2.89** | **2.38x** |
| Qwen2.5-VL 3B | Medusa | 0.57 | 1.07x | 0.60 | 1.12x | 0.66 | 1.08x | 0.59 | 1.12x | 0.62 | 1.21x | 0.60 | 1.16x | 0.65 | 1.21x | 0.61 | 1.15x | 0.61 | 1.14x |
| | EAGLE-2 | 1.18 | 1.41x | 1.03 | 1.30x | 0.98 | 1.26x | 1.07 | 1.38x | 1.40 | 1.60x | 1.11 | 1.32x | 1.39 | 1.52x | 1.11 | 1.32x | 1.16 | 1.39x |
| | ViSpec | **1.99** | **1.87x** | **2.13** | **1.81x** | **2.15** | **1.85x** | **1.96** | **1.82x** | **2.37** | **2.15x** | **2.22** | **1.71x** | **2.28** | **2.01x** | **2.37** | **1.78x** | **2.19** | **1.87x** |
| Qwen2.5-VL 7B | Medusa | 0.60 | 1.13x | 0.59 | 1.06x | 0.58 | 1.05x | 0.59 | 1.19x | 0.61 | 1.11x | 0.59 | 1.09x | 0.64 | 1.19x | 0.62 | 1.05x | 0.60 | 1.11x |
| | EAGLE-2 | 1.40 | 1.49x | 1.19 | 1.36x | 1.14 | 1.23x | 1.29 | 1.54x | 1.46 | 1.50x | 1.27 | 1.20x | 1.53 | 1.54x | 1.42 | 1.32x | 1.34 | 1.40x |
| | ViSpec | **2.19** | **1.84x** | **2.16** | **1.74x** | **2.21** | **1.72x** | **2.15** | **1.96x** | **2.27** | **1.99x** | **2.31** | **1.71x** | **2.30** | **1.91x** | **2.34** | **1.55x** | **2.24** | **1.80x** |
| | | | | | | | | | Temperature=1 | | | | | | | | | | |
| LLaVA-1.6 7B | Medusa | 0.58 | 1.36x | 0.58 | 1.37x | 0.57 | 1.32x | 0.56 | 1.35x | 0.58 | 1.67x | 0.57 | 1.29x | 0.60 | 1.19x | 0.59 | 1.32x | 0.58 | 1.36x |
| | EAGLE-2 | 1.78 | 2.17x | 0.51 | 1.34x | 0.41 | 1.11x | 1.02 | 1.53x | 1.03 | 1.78x | 0.77 | 1.32x | 1.33 | 1.47x | 0.98 | 1.57x | 0.98 | 1.54x |
| | ViSpec | **2.06** | **2.20x** | **1.94** | **1.99x** | **1.78** | **1.93x** | **1.96** | **1.98x** | **2.36** | **3.05x** | **2.32** | **2.21x** | **2.11** | **1.83x** | **2.16** | **1.94x** | **2.09** | **2.14x** |
| LLaVA-1.6 13B | Medusa | 0.68 | 1.41x | 0.67 | 1.44x | 0.66 | 1.42x | 0.66 | 1.40x | 0.67 | 1.40x | 0.64 | 1.37x | 0.70 | 1.37x | 0.68 | 1.37x | 0.67 | 1.40x |
| | EAGLE-2 | 1.51 | 1.98x | 1.29 | 1.73x | 1.26 | 1.72x | 1.45 | 1.78x | 1.54 | 1.83x | 1.46 | 1.72x | 1.64 | 1.73x | 1.60 | 1.79x | 1.47 | 1.79x |
| | ViSpec | **2.02** | **2.25x** | **1.98** | **2.15x** | **1.90** | **2.08x** | **2.07** | **2.08x** | **2.43** | **2.39x** | **2.04** | **2.01x** | **2.19** | **2.03x** | **2.22** | **2.07x** | **2.11** | **2.13x** |
| Qwen2.5-VL 3B | Medusa | 0.52 | 1.02x | 0.48 | 1.02x | 0.46 | 0.99x | 0.46 | 1.02x | 0.51 | 1.03x | 0.46 | 0.99x | 0.55 | 1.13x | 0.49 | 1.03x | 0.49 | 1.03x |
| | EAGLE-2 | 0.92 | 1.25x | 0.70 | 1.19x | 0.70 | 1.06x | 0.84 | 1.26x | 0.97 | 1.28x | 0.84 | 1.19x | 1.02 | 1.31x | 0.86 | 1.16x | 0.86 | 1.21x |
| | ViSpec | **1.49** | **1.49x** | **1.23** | **1.39x** | **1.32** | **1.38x** | **1.45** | **1.58x** | **1.42** | **1.50x** | **1.39** | **1.43x** | **1.49** | **1.59x** | **1.55** | **1.42x** | **1.42** | **1.47x** |
| Qwen2.5-VL 7B | Medusa | 0.56 | 1.05x | 0.51 | 0.95x | 0.49 | 0.96x | 0.51 | 1.02x | 0.52 | 1.00x | 0.50 | 1.02x | 0.53 | 1.02x | 0.53 | 1.02x | 0.52 | 1.01x |
| | EAGLE-2 | 1.19 | 1.52x | 0.92 | 1.19x | 0.88 | 1.08x | 1.00 | 1.23x | 1.08 | 1.22x | 0.94 | 1.13x | 1.11 | 1.32x | 1.04 | 1.19x | 1.02 | 1.18x |
| | ViSpec | **1.82** | **1.62x** | **1.57** | **1.47x** | **1.51** | **1.37x** | **1.61** | **1.49x** | **1.63** | **1.50x** | **1.88** | **1.53x** | **1.61** | **1.56x** | **1.70** | **1.38x** | **1.66** | **1.49x** |

table reports the average acceptance length $\tau$ and speedup ratios, calculated as the ratio of the average time required for standard autoregressive decoding to that of each method per token, under two temperature settings (0 and 1).

ViSpec consistently outperforms both Medusa and EAGLE-2 across all evaluated tasks and models, achieving the highest speedup ratios and $\tau$ values. For instance, at temperature 0 with LLaVA-v1.6-Vicuna-7B, ViSpec achieves a speedup of $2.90\times$ on TextVQA, surpassing EAGLE-2 ($1.25\times$) and Medusa ($1.46\times$) by a wide margin. Similarly, with LLaVA-v1.6-Vicuna-13B at temperature 0, ViSpec delivers a speedup of $2.57\times$ on ScienceQA, compared to EAGLE-2 ($2.12\times$) and Medusa ($1.61\times$). At temperature 1, ViSpec maintains its advantage, achieving a speedup of $2.25\times$ on ScienceQA with LLaVA-v1.6-Vicuna-13B, outperforming EAGLE-2 ($1.98\times$) and Medusa ($1.41\times$). These results underscore ViSpec's superior acceleration capabilities, with speedup ratios ranging from $1.37\times$ to $3.22\times$, compared to EAGLE-2 ($1.06\times$ to $2.17\times$) and Medusa ($0.95\times$ to $1.67\times$).

ViSpec demonstrates robust performance and generalizability across a diverse set of tasks, with notably high acceptance lengths on TextVQA, VizWiz, SEED-Bench, and COCO Captions. This suggests that its vision-aware approach effectively handles the varied sequential patterns inherent in these tasks. In contrast, the performance of EAGLE-2 and Medusa is more task-dependent. While they perform adequately on tasks like ScienceQA, they struggle on others, such as TextVQA and MM-Vet, particularly when compared to ViSpec. This indicates that their general-purpose draft mechanisms may not adapt as effectively to the complexities of visual-linguistic sequences.

Performance also varies across model architectures. LLaVA-1.6 models generally achieve higher speedup ratios and acceptance lengths compared to Qwen2.5-VL models. Such differences can be

attributed to the significantly larger vocabulary sizes of Qwen models, potentially increasing the complexity of token prediction.

## 5.3 Ablation Studies

**Impact of Compressed Image Embedding Count.** We evaluate the effect of varying the number of compressed image embeddings from 1 to 64 on ViSpec's performance, with results shown in Tab. 2. When the number remains significantly smaller than the original thousands of image embeddings, increasing the count has minimal impact on the average acceptance length $\tau$. However, it reduces the speedup ratio due to the increased computational load on the draft model during token generation. A single compressed image embedding adequately captures essential visual information, prompting us to adopt one compressed embedding in our final implementation.

Table 2: Impact of varying the number of compressed image embeddings on ViSpec's performance across three datasets, measured by average acceptance length $\tau$ and speedup ratio.

| Image Embeddings | COCO Captions | | GQA | | MME | |
|:---:|:---:|:---:|:---:|:---:|:---:|:---:|
| | $\tau$ | Speedup | $\tau$ | Speedup | $\tau$ | Speedup |
| 1 | 3.30 | 3.22x | 2.88 | 2.22x | 2.84 | 2.55x |
| 4 | 3.24 | 3.24x | 2.84 | 2.24x | 2.74 | 2.35x |
| 16 | 3.23 | 3.21x | 2.84 | 2.20x | 2.76 | 2.38x |
| 64 | 3.25 | 2.71x | 2.86 | 1.91x | 2.76 | 2.42x |

Table 3: Ablation study on the effectiveness of ViSpec's components across three datasets, measured by average acceptance length $\tau$ and speedup ratio, with EAGLE-2 as the baseline.

| Components | COCO Captions | | GQA | | MME | |
|:---:|:---:|:---:|:---:|:---:|:---:|:---:|
| | $\tau$ | Speedup | $\tau$ | Speedup | $\tau$ | Speedup |
| baseline | 1.24 | 1.80x | 1.74 | 1.64x | 1.25 | 1.68x |
| +image embedding compression | 2.04 | 2.37x | 2.15 | 1.92x | 2.04 | 1.83x |
| +global visual injection | 2.14 | 2.42x | 2.25 | 2.03x | 2.14 | 1.95x |
| +dataset generation | 3.30 | 3.22x | 2.88 | 2.22x | 2.84 | 2.55x |

**Effectiveness of Each Component.** We conduct an ablation study to assess the contribution of ViSpec's core components: image embedding compression, global visual feature injection, and dataset generation. Using EAGLE-2 [18] as the baseline, we report the average acceptance length and speedup ratio across the COCO Captions, GQA, and MME datasets, as shown in Tab. 3. Adding image embedding compression increases the speedup ratio by up to 30%, enabling the draft model to efficiently process visual information. Incorporating global visual feature injection further improves speedup by 7%, underscoring its role in maintaining persistent visual context and enhancing multimodal coherence. The inclusion of dataset generation yields an additional 30% speedup, equipping the draft model to handle extended multimodal sequences effectively. Together, these components synergistically enhance ViSpec's acceleration performance while ensuring robust performance across diverse tasks.

**Vision Adaptor Overheads.** While the vision adaptor increases the draft model's parameter count, it theoretically reduces the prefill computation by processing fewer visual tokens. However, as draft models are small and efficient, we observe no statistically significant change in prefill latency (Tab. 4). The minor variations recorded are attributed to measurement noise.

**Output Length vs. Speedup.** Table 5 illustrates the relationship between the average output length and the achieved end-to-end speedup across various datasets. As expected, longer generation sequences generally yield higher speedup ratios, since they offer more opportunities for successful draft model predictions. Despite this trend, our method demonstrates robust performance, providing significant acceleration even on datasets characterized by shorter responses.

Table 4: Analysis of vision adaptor overheads during the prefill stage on the COCO Captions dataset.

| Model | Params (M) | | GFLOPs | | Latency (s) | |
|---|---|---|---|---|---|---|
| | Base | +Adap | Base | +Adap | Base | +Adap |
| LLaVA-1.6 7B | 367 | 451 | 956 | 179 | 0.227 | 0.231 |
| LLaVA-1.6 13B | 534 | 665 | 1460 | 279 | 0.334 | 0.334 |
| Qwen2.5-VL 3B | 404 | 425 | 57.3 | 18.3 | 0.002 | 0.004 |
| Qwen2.5-VL 7B | 826 | 890 | 172 | 55.5 | 0.018 | 0.016 |

Table 5: Relationship between output length and speedup ratio.

| Dataset | Tokens | Speedup |
|---|---|---|
| GQA | 46.25 | 2.22x |
| SEED-Bench | 57.66 | 2.22x |
| SQA | 74.07 | 2.37x |
| VizWiz | 105.91 | 2.26x |
| MME | 115.01 | 2.55x |
| MM-Vet | 171.13 | 2.52x |
| COCO Caps | 236.04 | 3.21x |
| TextVQA | 353.58 | 2.90x |

# 6    Conclusion

We introduce **Vision-Aware Speculative Decoding (ViSpec)**, the first framework to achieve significant acceleration for vision-language models (VLMs) through speculative decoding. By integrating compressed image embeddings, persistent global visual feature injection, and synthetic long-response dataset generation, ViSpec addresses key limitations in processing multimodal sequences with shallow draft models. Our experiments demonstrate speedups of up to $3.22\times$ across diverse VLMs and tasks, establishing ViSpec as a pioneering solution for multimodal inference acceleration. Despite this breakthrough, ViSpec's absolute speedup trails state-of-the-art text-only methods. We identify two primary avenues for improvement: first, curating higher-quality multimodal training datasets with greater conversational depth to enhance the draft model's predictive accuracy; second, optimizing vision encoder architectures, potentially via dynamic patch reduction or neural compression, to reduce visual processing overhead. These advancements, coupled with hardware-aware kernel optimizations, could bridge the performance gap between multimodal and text-only speculative decoding, enabling real-time deployment of advanced VLMs.

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

# A  Implementation Details

**Vanilla.** We utilize models from the Hugging Face Transformers library with the PyTorch backend and pre-allocated KV cache. All other methods build upon these models.

**Medusa.** We implement a 1-layer, 3-head Medusa model, adhering to its default configuration. For training on both text-only and vision-language datasets, we use a learning rate of 3e-5, a batch size of 8, and the AdamW optimizer. The model is trained for 20 epochs with a 1-epoch warmup and linear learning rate decay. We set a maximum sequence length of 2048 for both dataset types. For inference, we adopt EAGLE-2's draft tree structure, configuring a total of 30 draft tokens, a tree depth of 3, and selecting 8 nodes during the expansion phase across all models and tasks.

**EAGLE-2.** We employ a 1-layer EAGLE-2 model, following its default settings. Training on text-only and vision-language datasets uses a learning rate of 3e-5, a batch size of 8, and the AdamW optimizer. The model is trained for 20 epochs with a 1-epoch warmup and linear learning rate decay, with a maximum sequence length of 2048 for both dataset types. For inference, we use EAGLE-2's draft tree with 30 draft tokens, a tree depth of 3, and 8 nodes selected during expansion, applied uniformly across all models and tasks.

**ViSpec.** We implement a single-layer draft model that mirrors a decoder layer of the target model. For training on text-only and vision-language datasets, we use a learning rate of 3e-6, a batch size of 8, and the AdamW optimizer. The model is trained for 20 epochs with a 1-epoch warmup and linear learning rate decay, supporting a maximum sequence length of 2048 for both dataset types. During inference, we adopt EAGLE-2's draft tree structure, configuring 30 draft tokens, a tree depth of 3, and selecting 8 nodes during expansion, applied consistently across all models and tasks.

**Generation Prompts.** For training dataset generation, we append the prompt "*Please answer with at least 1000 words.*" to each sample to elicit long responses. For inference, we use task-specific prompts to encourage detailed responses. For visual question answering (VQA) tasks, the prompt is: "*Please answer with an explanation.*" For optical character recognition (OCR) tasks, the prompt is: "*Perform an OCR task on the provided image. Extract the text accurately and provide a detailed explanation of the process. Ensure the response is comprehensive and well-structured.*" For captioning tasks, the prompt is: "*Provide a detailed description of the given image.*" For ScienceQA, we use its official chain-of-thought prompt to generate the answer, followed by the lecture and explanation (QCM→ALE).

# B  Additional Experiments

## B.1  Experiments on High-Resolution Datasets

We conducted experiments on high-resolution datasets [31, 37], where ViSpec continues to demonstrate strong performance. Table 6 compares ViSpec against the EAGLE-2 baseline using both LLaVA-1.6 7B and Qwen2.5-VL 7B.

Table 6: Performance on high-resolution datasets, comparing average acceptance length $\tau$ and speedup ratio.

| Model | Dataset | Method | $\tau$ | Speedup |
|---|---|---|---|---|
| LLaVA-1.6 7B | HR-Bench 4K | EAGLE-2 | 1.43 | 1.52x |
| | | ViSpec | **2.86** | **1.93x** |
| | MME-RealWorld | EAGLE-2 | 1.42 | 1.75x |
| | | ViSpec | **2.85** | **2.35x** |
| Qwen2.5-VL 7B | HR-Bench 4K | EAGLE-2 | 0.34 | 0.90x |
| | | ViSpec | **2.16** | **1.29x** |
| | MME-RealWorld | EAGLE-2 | 0.52 | 0.95x |
| | | ViSpec | **2.11** | **1.37x** |

Notably, Qwen-VL does not cap its input image token count, resulting in a longer prefill time for high-resolution datasets. Since speculative decoding accelerates only the decoding stage, this extended

prefill duration reduces the overall speedup ratio. However, ViSpec's robust average acceptance length $\tau$ indicates that the decoding phase itself is still effectively accelerated.

## B.2 Experiments with Temporal Data

In principle, ViSpec could be more effective for video inputs, as videos contain temporal redundancy in addition to the spatial redundancy found in static images. From an input processing standpoint, this task is not fundamentally different from handling image patches, as video inputs are typically processed as a sequence of frame embeddings. To test this hypothesis, we apply our draft model, which was trained exclusively on static image data, directly to video tasks without fine-tuning.

For this preliminary experiment, we compress each video frame into a single embedding, average their global features, and evaluate the Qwen2.5-VL 7B model on the MSVD-QA [6] and MVBench [17] datasets. MSVD-QA is a video question-answering task, while MVBench is a benchmark evaluating temporal understanding across 20 different tasks. We limit the input frames, as processing more would lengthen the prefill time, thereby reducing the speedup gained from the accelerated decoding stage. The results are presented in Tab. 7.

Table 7: Performance of Qwen2.5-VL 7B on video datasets, comparing average acceptance length $\tau$ and speedup ratio.

| Dataset | Method | $\tau$ | Speedup |
|---------|--------|--------|---------|
| MSVD-QA | EAGLE-2 | 1.10 | 1.22x |
|         | ViSpec | **2.16** | **1.46x** |
| MVBench | EAGLE-2 | 0.83 | 0.83x |
|         | ViSpec | **2.09** | **1.32x** |

The results demonstrate that ViSpec achieves a notable speedup even without video-specific training. Developing a dedicated framework optimized for video data remains a promising direction for future work.

