# OpenReview forum: "ViSpec: Accelerating Vision-Language Models with Vision-Aware Speculative Decoding"
_NeurIPS.cc/2025/Conference — NeurIPS 2025 poster_

### Official Review · Reviewer_3Mw6 · 2025-06-13

**Clarity:** 3
**Significance:** 4
**Originality:** 4
**Rating:** 5
**Confidence:** 4

**Summary:**

This paper introduces a novel framework to accelerate inference in VLMs. The core idea is that small draft models cannot efficiently process the highly redundant visual information that large target models can. ViSpec addresses this by using a lightweight vision adaptor to create a compact image representation for the draft model and a global feature augmentation technique to maintain multimodal coherence. The paper  also creates a new dataset with long-form responses to facilitate training. ViSpec achieves significant speedups on various benchmarks.

**Questions:**

See the weaknesses above.

**Ethical Concerns:**

["NO or VERY MINOR ethics concerns only"]

**Final Justification:**

The rebuttal has addressed my concerns about the potential overheads and generality of the proposed method, and it demonstrates superior performance compared to previous work. I agree with the other reviewers that the work is designed specifically to address the unique challenges of applying speculative decoding for VLMs through thorough theoretical analysis and valid experiments, which provide an interesting perspective for accelerating the decoding phase of VLMs. Overall, I would like to raise my evaluation score.

**Limitations:**

The authors have addressed limitations and social impact of the work.

**Quality:**

3

**Strengths And Weaknesses:**

Strengths And Weaknesses

Strengths:

1. The proposed method is technically sound and well-motivated. The design of the vision adaptor is a simple but effective solution tailored for the speculative decoding for vision language models.

2. The analysis that speculative decoding for LLM is suboptimal for VLMs is insightful.

3. The experiment results are competitive. The method demonstrates advantages over previous SOTA methods on multiple VLM benchmarks.

4. The implementation details and experiment settings are clearly described.

Weaknesses:

1. The paper lacks comparison with existing speculative decoding methods for VLMs, such as [1] and [2].

2. The claim that the vision adaptor is "lightweight" is not substantiated with a quantitative analysis. A detailed breakdown of the associated overheads is necessary.

3. The authors should investigate how the speedup varies with the length of the output, as it is critical for understanding the method's boundary.

4. To demonstrate broader utility, how about the performance of the method on text-only inputs?  It is a frequent use case for VLMs.

[1] Gagrani, Mukul, et al. "On speculative decoding for multimodal large language models."

[2] Lee, Minjae, et al. "In-batch Ensemble Drafting: Toward Fast and Robust Speculative Decoding for Multimodal Language Models."

---

> ### Author Rebuttal · Authors · 2025-07-30
>
> We are grateful for your insightful questions and suggestions for improvement.
>
> ### 1. Comparison with existing speculative decoding methods for VLMs
>
> We acknowledge and have cited these valuable prior works in our paper. However, as their source code is not publicly available, we encountered difficulties in reproducing their results for a direct, controlled comparison. Therefore, we directly cite the performance data reported in their original papers and apply our method to the same LLaVA-1.5 7B model and datasets they used to ensure a fair comparison. The results are as follows:
>
> |       Method       |  SQA  | COCO Caption | TextVQA | VQAv2 |
> | :----------------: | :---: | :----------: | :-----: | :---: |
> | Gagrani et al. [1] | 1.46x |    1.37x     |    -    |   -   |
> |   Lee et al. [2]   |   -   |      -       |  1.61x  | 1.72x |
> |   ViSpec (ours)    | 2.48x |    3.25x     |  2.67x  | 2.53x |
>
>
> As shown in the table, our method significantly outperforms the previous state-of-the-art works across all common benchmarks.
>
> ### 2. Detailed breakdown of vision adaptor overheads
>
> Theoretically, the image adaptor increases the parameter count of the draft model but reduces its computational load during prefilling by processing fewer tokens. Since the draft model is already very small and efficient, we observed no statistically significant impact on its overall speed. The analysis of the prefilling stage on the COCO Captions dataset is presented below:
>
> |     Model     | Parameters (M) | Parameters (M) |   GFLOPS    |   GFLOPS   | Prefill Latency (s) | Prefill Latency (s) |
> | :-----------: | :------------: | :------------: | :---------: | :--------: | :-----------------: | :-----------------: |
> |               |  w/o Adaptor   |   w/ Adaptor   | w/o Adaptor | w/ Adaptor |     w/o Adaptor     |     w/ Adaptor      |
> | LLaVA-1.6 7B  |      367       |      451       |     956     |    179     |        0.227        |        0.231        |
> | LLaVA-1.6 13B |      534       |      665       |    1460     |    279     |        0.334        |        0.334        |
> |  Qwen-VL 3B   |      404       |      425       |    57.3     |    18.3    |        0.002        |        0.004        |
> |  Qwen-VL 7B   |      826       |      890       |     172     |    55.5    |        0.018        |        0.016        |
>
>
> And the token rate for the decoding stage:
>
> |     Model     |     ms/token      |     ms/token     |
> | :-----------: | :---------------: | :--------------: |
> |               | w/o Image Adaptor | w/ Image Adaptor |
> | LLaVA-1.6 7B  |       1.077       |      1.185       |
> | LLaVA-1.6 13B |       1.225       |      1.192       |
> |  Qwen-VL 3B   |       1.081       |      1.211       |
> |  Qwen-VL 7B   |       1.402       |      1.539       |
>
>
> We attribute these minor speed variations primarily to measurement noise.
>
> ### 3. Relationship between output length and speedup ratio
>
> The VLMs we tested tend to generate text of varying lengths across different datasets. We present the relationship between the average number of generated tokens and the speedup below:
>
> |  Dataset   | Average New Tokens | Speedup Ratio |
> | :--------: | :----------------: | :-----------: |
> |    GQA     |       46.25        |     2.22x     |
> | SEED-Bench |       57.66        |     2.22x     |
> |    SQA     |       74.07        |     2.37x     |
> |   VizWiz   |       105.91       |     2.26x     |
> |    MME     |       115.01       |     2.55x     |
> |   MM-Vet   |       171.13       |     2.52x     |
> | COCO Caps  |       236.04       |     3.21x     |
> |  TextVQA   |       353.58       |     2.90x     |
>
>
> Generally, longer text outputs tend to yield a higher speedup ratio, as there are more opportunities for the draft model to make correct predictions. Nevertheless, our method maintains stable and robust performance across all datasets, even those with shorter response lengths.
>
> ### 4. Performance on text-only inputs
>
> We evaluate the speedup on the text-only **MT-Bench** [3] dataset using the LLaVA-1.6 7B model. We compare the performance of the original text-only draft model with our draft model that has been fine-tuned on multimodal data. The speedup after our multimodal fine-tuning is **0.96x** of the speedup achieved before fine-tuning. This demonstrates that our proposed method has a negligible impact on performance for text-only tasks, preserving the model's utility as a general-purpose assistant.
>
> [1] M. Gagrani, et al. "On speculative decoding for multimodal large language models." 2024.
>
> [2] M. Lee, et al. "In-batch Ensemble Drafting: Toward Fast and Robust Speculative Decoding for Multimodal Language Models." 2024.
>
> [3] L. Zheng, et al. "Judging LLM-as-a-Judge with MT-Bench and Chatbot Arena." NeurIPS, 2023.

---

### Official Review · Reviewer_mKSj · 2025-06-25

**Clarity:** 3
**Significance:** 2
**Originality:** 2
**Rating:** 4
**Confidence:** 3

**Summary:**

This paper introduces ViSpec, aiming to achieve acceleration for vision-language models (VLMs) through speculative decoding. ViSpec employs a lightweight vision adapter module to compress image tokens into a compact representation. It extracts a global feature vector for each input image and augments all subsequent text tokens with this feature to enhance multimodal coherence. Experiments show that the proposed method achieves significant speedup.

**Questions:**

Why report only the speed-up metrics and not the corresponding accuracy?

**Ethical Concerns:**

["NO or VERY MINOR ethics concerns only"]

**Final Justification:**

Most of my concerns have been addressed. Therefore, I would like to raise my scores.

**Limitations:**

Yes

**Paper Formatting Concerns:**

No major formatting issues.

**Quality:**

2

**Strengths And Weaknesses:**

Strengths:

ViSpec integrates compressed image embeddings, persistent global visual feature injection, and synthetic long-response dataset generation to address the limitations in processing multimodal sequences with shallow draft models. ViSpec achieves significant speedup compared to previous methods.

Weaknesses

1.	The paper only reports speedup metrics but not performance metrics. What is the performance of the model after acceleration?
2.	High-resolution images require more image tokens and thus necessitate acceleration. It would be beneficial to evaluate the acceleration performance on benchmarks containing high-resolution images, such as HR-Bench benchmark and MME-Realworld benchmark.
3.	It lacks comparisons with relevant methods, such as FastV[1] which compresses using attention weights and MiniCPM-V[2] which employs learnable queries for compression.

[1] Chen L, Zhao H, Liu T, et al. An image is worth 1/2 tokens after layer 2: Plug-and-play inference acceleration for large vision-language models[C]//European Conference on Computer Vision. Cham: Springer Nature Switzerland, 2024: 19-35.

[2] Yao Y, Yu T, Zhang A, et al. Minicpm-v: A gpt-4v level mllm on your phone[J]. arXiv preprint arXiv:2408.01800, 2024.

---

> ### Author Rebuttal · Authors · 2025-07-30
>
> We sincerely thank you for your insightful feedback and the opportunity to clarify these important aspects of our work.
>
> ### 1. Only reporting speedup metrics
>
> Our work, ViSpec, is based on the principle of **speculative decoding**. The token compression and drafting process *ONLY* involves a small, auxiliary **draft model**. The final output is always verified by the original, unmodified **target model**. The speculative decoding process has been mathematically proven to be **lossless**, meaning it produces an output distribution that is identical to that of standard autoregressive decoding from the target model alone [1, 2].
>
> Because the final output is provably identical, the model's accuracy and any other performance metrics remain unchanged. Consequently, it is standard practice in the speculative decoding literature to report only the acceleration (i.e., speedup), as performance metrics are not affected. Our paper follows this established convention.
>
> ### 2. Experiment on benchmarks containing high-resolution images
>
> We conduct experiments on the suggested benchmarks, and our method continues to demonstrate strong performance. The results are presented below, comparing ViSpec against the EAGLE-2 baseline for both LLaVA-1.6 7B and Qwen2.5-VL 7B.
>
> | Model         | Benchmark     | Method  | Mean Acceptance Length | Speedup Ratio |
> | :-----------: | :-----------: | :-----: | :--------------------: | :-----------: |
> | LLaVA-1.6 7B  | HR-Bench      | ViSpec  | 2.86                   | 1.93x         |
> |               |               | EAGLE-2 | 1.43                   | 1.52x         |
> |               | MME-Realworld | ViSpec  | 2.85                   | 2.35x         |
> |               |               | EAGLE-2 | 1.42                   | 1.75x         |
> | Qwen2.5-VL 7B | HR-Bench      | ViSpec  | 2.16                   | 1.29x         |
> |               |               | EAGLE-2 | 0.34                   | 0.90x         |
> |               | MME-Realworld | ViSpec  | 2.11                   | 1.37x         |
> |               |               | EAGLE-2 | 0.52                   | 0.95x         |
>
>
> Notably, Qwen-VL does not limit its input image token count, leading to a significantly longer prefilling time on high-resolution benchmarks. Since speculative decoding accelerates only the decoding stage, this extended prefilling time results in a lower overall speedup ratio. However, ViSpec’s mean acceptance length remains robust, indicating that the acceleration of the decoding phase itself is effective. Interestingly, the baseline method actually decelerates the Qwen-VL model on these benchmarks (i.e., speedup < 1.0x). This finding further validates our core argument from Section 4.1: a shallow draft model’s performance inherently degrades when processing the long and redundant token sequences from high-resolution images, highlighting the critical need for the compression strategy that ViSpec introduces. We will include this analysis in the final version of our paper.
>
> ### 3. Comparison with FastV and MiniCPM-V
>
> Our method is fundamentally different from approaches like FastV and MiniCPM-V. The core distinction lies in the nature of the acceleration:
>
> * **FastV & MiniCPM-V (Lossy Compression & Prefill Focus)**: These methods are **lossy model compression** techniques. Their main goal is to accelerate the computationally intensive **prefilling stage** by processing fewer visual tokens. For instance, FastV reduces the number of tokens fed into the transformer, which cuts down the FLOPs for prefilling. However, this approach offers minimal benefit for the **decoding stage**. Autoregressive decoding is typically **memory-bound**, where the bottleneck is loading model weights from memory for each forward pass, not the computation itself. Since these methods still require the full model weights to be loaded for every single token generation, they do not alleviate this memory bandwidth bottleneck, and the actual wall-clock speedup during decoding is often negligible.
>
> * **ViSpec (Lossless Acceleration & Decode Focus)**: Our method is a speculative decoding framework that accelerates an existing, unmodified target model. It is mathematically guaranteed to be **lossless**, meaning it does not alter the model's final output quality in any way. Our focus is purely on reducing the **decoding latency**. In stark contrast to FastV & MiniCPM-V, ViSpec is designed specifically to accelerate the memory-bound decoding phase by reducing the number of sequential steps required to generate the full output.
>
> Our method is, in fact, **orthogonal** to approaches like FastV and MiniCPM-V. FastV or MiniCPM-V can be used to compress the target model itself, accelerating the computationally-intensive **prefill stage**. Following that, our ViSpec framework can be applied to this already-optimized model to accelerate its memory-bound **decoding stage**. Therefore, these methods are not in conflict; combining them could lead to a cumulative effect, further enhancing overall inference efficiency by optimizing both prefill and decoding stages.
>
> [1] Y. Leviathan, et al. "Fast Inference from Transformers via Speculative Decoding." ICML Oral, 2023.
>
> [2] C. Chen, et al. "Accelerating Large Language Model Decoding with Speculative Sampling." DeepMind, 2023.

---

> > ### Comment · Reviewer_mKSj · 2025-08-05
> > **Response to authors**
> >
> > Thank the authors for their detailed rebuttal. Most of my concerns have been addressed.

---

### Official Review · Reviewer_DJLc · 2025-07-03

**Clarity:** 3
**Significance:** 3
**Originality:** 3
**Rating:** 2
**Confidence:** 5

**Summary:**

- This paper introduces Vision-Aware Speculative Decoding (ViSpec) to achieve acceleration for vision-language models (VLMs) through speculative decoding.

- By integrating compressed image embeddings, persistent global visual feature injection, and synthetic long-response dataset generation, ViSpec addresses key limitations in processing multimodal sequences with shallow draft models.

- The experiments demonstrate speedups across diverse VLMs and tasks.

- This paper identifies two primary avenues for improvement: first, curating higher-quality multimodal training datasets with greater conversational depth to enhance the draft model’s predictive accuracy; second, optimizing vision encoder architectures, potentially via dynamic patch reduction or neural compression, to reduce visual processing overhead.

**Questions:**

- Most multimodal benchmarks only require a single-word response, such as 'yes' or 'no' for MME and 'A', 'B', 'C', 'D' for multi-choice QA.
Thus, it is unclear why there are more than 2x-3x speedups for ViSpec in those benchmarks.

- Lack of experiments in high-concurrency settings. The throughput for large batch sizes is unknown.

**Ethical Concerns:**

["NO or VERY MINOR ethics concerns only"]

**Final Justification:**

Thanks for your response. I have read the rebuttal and other reviews. While I appreciate your response, I would like to keep my scores.

**Limitations:**

yes.

**Paper Formatting Concerns:**

None.

**Quality:**

3

**Strengths And Weaknesses:**

Strengths

- This paper hypothesizes that large VLMs can effectively filter redundant image information layer by layer without compromising textual comprehension, whereas smaller draft models struggle to do so.

- This paper introduces Vision-Aware Speculative Decoding (ViSpec). ViSpec employs a lightweight vision adaptor module to compress image tokens into a compact representation.

- This work extracts a global feature vector for each input image and augments all subsequent text tokens with this feature to enhance multimodal coherence.

- This work curates a specialized training dataset by repurposing existing datasets and generating extended outputs using the target VLM with modified prompts.

Weaknesses

- The vision adaptor is heavily inspired by BLIP-2's Q-Former, and the global feature injection resembles EAGLE's target-aware mechanisms.
The incremental combination isn't sufficiently differentiated from prior work.

- High speedups on COCO Captions but low speedup on TextVQA. The paper doesn't analyze the inconsistent gains.

- Recent VLMs handle video and 3D input, but ViSpec is only evaluated on static images. Compression may fail for temporal data.

---

> ### Author Rebuttal · Authors · 2025-07-30
>
> We appreciate you taking the time to review our paper and providing this valuable feedback. We address your specific concerns below.
>
> ### 1. Novelty of this work against BLIP-2 and EAGLE
>
> Our primary contribution is the identification and analysis of a fundamental bottleneck in **applying speculative decoding to Vision-Language Models**, an issue *unaddressed* by prior work like **BLIP-2 and EAGLE**. We subsequently introduce a comprehensive solution to tackle this challenge: a lightweight Vision Adapter to compress redundant visual information and a Global Feature Injection mechanism to combat the *lost-in-the-middle* effect. These are complemented by novel dataset generation and training strategies designed to produce high-quality, long-form training data and to train a robust, effective draft model.
>
> In **Section 4.1**, we provide a theoretical analysis from the unique perspective of speculative decoding. We demonstrate how the performance of a **shallow draft model** degrades when processing the long and highly redundant sequences typical of image patch embeddings. Our analysis reveals that as the number of redundant visual tokens increases, a shallow Transformer's attention mechanism becomes saturated, effectively averaging over the redundant inputs while ignoring unique and critical information. This leads to our central argument: **for speculative decoding to be effective in a multimodal context, this redundant visual information should be compressed before it is processed by the draft model**. This core insight directly motivates our structural solution: the vision adaptor for compression and the global feature injection for maintaining context.
>
> ### 2. Performance gap between COCO Captions and TextVQA
>
> The difference in speedup between COCO Captions (3.22x) and TextVQA (2.90x) likely stems from the differing granularity of these two tasks. COCO Captions require a holistic, overall description of the image, a task for which our compressed visual representation is highly effective for the draft model. In contrast, TextVQA is a fine-grained task that demands focusing on specific text within the image, token compression may reduce the draft model's ability to predict these fine-grained details, lowering the acceptance rate. This does not affect the final output quality, as the target model makes corrections, but it does logically result in a lower speedup ratio.
>
> Another factor that can contribute to varying speedups across different benchmarks is the length of the generated output. This occurs because speculative decoding accelerates only the **decoding** stage, not the **prefilling** stage. Since the overall speedup calculation includes the fixed prefilling time, tasks that produce longer sequences see a greater impact from acceleration. For instance, on the GQA dataset, where the average response is short (46 tokens), we observe a 2.22x speedup. In contrast, on MME, with longer responses (115 tokens), the speedup increases to 2.55x. For longer outputs, the accelerated decoding phase constitutes a larger proportion of the total inference time, naturally leading to a higher end-to-end speedup ratio.
>
> ### 3. Experiment with Temporal Data
>
> In principle, our method could be even *more* effective for video inputs, as video data contains additional temporal redundancy compared with static images. Since video inputs are typically processed as a sequence of embeddings, the problem is not fundamentally different from handling image patches. To test this hypothesis, we apply our draft model, which was trained purely on static image data, directly to video tasks. For a preliminary experiment, we compress each video frame into a single embedding, average their corresponding global features, and evaluate the Qwen2.5-VL 7B model on the MSVD-QA [1] and MVBench [2] datasets. The MSVD-QA dataset is a video question-answering task where the model must answer natural language questions based on the content of a video. MVBench is a benchmark specifically designed to evaluate temporal understanding in videos, featuring 20 different tasks that require reasoning about actions, their sequence, and temporal relationships. We limit the maximum number of frames to 32, as processing more frames would only lengthen the prefilling time, diminishing the speedup gained from accelerating the decoding stage. Results are presented below:
>
> | Benchmark | Method  | Mean Acceptance Length | Speedup Ratio |
> | :-------: | :-----: | :--------------------: | :-----------: |
> | MSVD-QA   | ViSpec  | 2.16                   | 1.46x         |
> |           | EAGLE-2 | 1.10                   | 1.22x         |
> | MVBench   | ViSpec  | 2.09                   | 1.32x         |
> |           | EAGLE-2 | 0.83                   | 0.83x         |
>
>
> The results demonstrate that our ViSpec method achieves a notable speedup even without any video-specific training, whereas the baseline method actually **decelerates** inference on MVBench (0.83x speed). Developing a dedicated framework optimized for video remains a promising direction for future work.
>
> ### 4. Speedup on multimodal benchmarks such as MME
>
> As we have described in **Appendix A, "Implementation Details,"** following [3], we use modified prompts to instruct the model to generate long-form, explanatory answers, rather than just single-word responses. For example, we ask the model to "*Provide a detailed description of the given image.*" This allows us to rigorously evaluate the decoding acceleration on these benchmarks.
>
> ### 5. Experiment in high-concurrency settings
>
> We conduct experiments on the LLaVA-1.6 7B model using the SQA dataset, with the batch size varying from 1 to 8. We compare ViSpec to the standard batched decoding inference in the Hugging Face Transformers library. The results are as follows:
>
> | Batch Size | Speedup Ratio |
> | :--------: | :-----------: |
> | 1          | 2.37x         |
> | 2          | 2.00x         |
> | 4          | 1.75x         |
> | 8          | 1.56x         |
>
>
> We still observe a substantial gain when using batched decoding. The principle of speculative decoding is to lower the number of memory access cycles to accelerate memory-bound decoding. Therefore, it is expected that the *relative* speedup over standard decoding will decrease as the batch size increases, since higher concurrency better utilizes the computation units. Nevertheless, our method maintains a significant speedup across all tested batch sizes. Implementation optimizations could further boost this performance.
>
>
> [1] D. Xu, et al. "Video question answering via gradually refined attention over appearance and motion." ACM-MM, 2017.
>
> [2] K. Li, et al. "MVBench: A comprehensive multi-modal video understanding benchmark." CVPR, 2024.
>
> [3] Gagrani, Mukul, et al. "On speculative decoding for multimodal large language models." 2024.

---

> > ### Comment · Reviewer_DJLc · 2025-08-07
> >
> > Thanks for your response. I have read the rebuttal and other reviews. While I appreciate your response, I would like to keep my scores.

---

### Official Review · Reviewer_3b1n · 2025-07-03

**Clarity:** 3
**Significance:** 2
**Originality:** 2
**Rating:** 4
**Confidence:** 3

**Summary:**

This paper proposes the ViSpec framework to address the slow inference speed of vision-language models (VLMs). It accelerates inference through a specially designed vision-aware speculative decoding technique.
ViSpec employs a lightweight visual adapter to compress image information and injects global visual features to maintain multimodal coherence. This design thereby tackles the challenge of redundant visual inputs that small draft models struggle to handle.
Experiments demonstrate that ViSpec achieves speedup on various VLMs without compromising much generation quality.

**Questions:**

Please polish the writing and complemente more analysis and experiments.

**Ethical Concerns:**

["NO or VERY MINOR ethics concerns only"]

**Final Justification:**

The authors have clarified my concerns, and the reviewer is willing to raise the score to 4.

**Limitations:**

Yes

**Quality:**

2

**Strengths And Weaknesses:**

Strengths
- This framework is specifically designed to address the unique challenges of VLMs. It handles redundant visual information through image embedding compression.

- This paper is easy to follow. Experiments show that ViSpec consistently outperforms baseline methods such as Medusa and EAGLE-2 across all tested models and tasks. ViSpec demonstrates strong effectiveness on four mainstream VLMs and eight different multimodal benchmarks.

- ViSpec introduces an innovative data generation approach to overcome the data limitation.

Weaknesses
- The "experimental setup" section is poorly written, with low information density and inappropriate sectioning.

- The "Global Visual Feature Integration" is not novel; a similar idea already exists in "ContextNet: Improving Convolutional Neural Networks for Automatic Speech Recognition with Global Context". The difference exists in different backbones.

- The paper claims that using only one compressed image embedding is sufficient to capture the necessary visual information but does not provide adequate justification. The compression might lead to the inability to recognize fine-grained small objects, but the paper did not test corresponding ability.

---

> ### Author Rebuttal · Authors · 2025-07-30
>
> We sincerely thank you for your detailed feedback and constructive suggestions. We address your points below.
>
> ### 1. Writing and sectioning issues
>
> We will refine the writing and consolidate these subsections in the final version. If you have more specific suggestions for improvement, we would be very grateful to hear them.
>
> ### 2. Architecturally analogous “Global Visual Feature Integration”
>
> The concept of a "global feature" is widely applied in deep learning. Its core idea is not only adopted in *ContextNet* [1] but also well-established in numerous other influential works, such as *Squeeze-and-Excitation Networks* for channel-wise feature recalibration [2], the `[CLS]` token in *BERT* for sentence-level representation [3], and the class token in *Vision Transformers* for image-level classification [4]. However, the mere presence of a global feature is not where the novelty lies. The crucial aspects are the specific **motivation** for its use and more importantly the **method** by which it is generated and integrated. In this regard, our Global Visual Feature Integration differs significantly from ContextNet.
>
> * **Motivation**:
>     ContextNet introduces the global context vector to address the limited receptive field of CNNs, which struggle with long-range dependencies in speech signals. Authors of ContextNet note that this is *NOT* considered a problem for architectures like RNNs or Transformers. In contrast, as we argue theoretically in Section 4.1, the core issue we address is the inherent deficiency of *shallow* Transformers (draft models) in processing long and redundant sequences, particularly a large number of image tokens. To the best of our knowledge, we are the first to identify this specific challenge. This insight leads to our proposal to compress image tokens. Consequently, the reduced number of image tokens relative to text tokens introduces the risk of the draft model "forgetting" the visual context. The Global Visual Feature Integration is therefore needed to mitigate this risk.
>
> * **Method**:
>     ContextNet generates a global context vector from a convolutional block's output using a Squeeze-and-Excitation (SE) module. This vector is then used to perform feature recalibration by multiplicatively rescaling the local feature maps. The global feature, therefore, modulates the local information. We derive the global visual feature vector directly from our vision adaptor module, which uses learnable queries to summarize the image. We then integrate this feature by additively injecting it into the hidden state of every subsequent text token via a learned projection. This ensures that a constant, global representation of the image is available at every step of the generation, acting as a continuous anchor to the visual input.
>
> In summary, while both approaches use a "global feature," they do so for different reasons (CNN limitations vs. draft model *forgetting*) and with different mechanisms (SE module vs. vision adaptor) tailored to their unique problems. Moreover, as we demonstrate in our ablation study in Section 5.3, our framework significantly outperforms prior work even **without** Global Visual Feature Integration, highlighting the contribution of our vision-aware token compression strategy.
>
> ### 3. Recognizing fine-grained objects with compressed embeddings
>
> A key principle of speculative decoding is that the performance of the draft model affects only the **acceleration speed**, *NOT* the **final output quality**. If the draft model fails to recognize a fine-grained object, its prediction is simply rejected and corrected by the target model. The output distribution is guaranteed to be identical to that of the target model decoding alone.
>
> In Table 2 of our main paper, we have already presented experiments on three diverse datasets (COCO Captions, GQA, and MME), varying the number of compressed image embeddings from 1 to 64. These results consistently show that increasing the number of embeddings beyond 1 does not yield meaningful gains in speedup, which demonstrates that a single compressed image embedding is often sufficient for answering common visual questions, including relatively fine-grained ones such as, “What is the short person holding?” (an example from GQA) .
>
> To further address the reviewer’s concern about fine-grained tasks, we conduct additional experiments on the more fine-grained OCR dataset *SynthDoG EN* [5]. The results below show the performance of LLaVA-1.6 7B with greedy decoding as we vary the number of compressed image embeddings.
>
> | Compressed Image Embeddings | Mean Acceptance Length | Speedup Ratio |
> | :-------------------------: | :--------------------: | :-----------: |
> |              1              |          1.98          |     1.97x     |
> |              4              |          2.02          |     1.93x     |
> |             16              |          2.00          |     1.92x     |
> |             64              |          1.99          |     1.90x     |
> |              -              |          1.05          |     1.45x     |
>
> We observe a performance drop on this benchmark compared to less fine-grained tasks like GQA (from 2.22x down to 1.97x), but our method still significantly outperforms the EAGLE-2 baseline, which uses no compression (last row). While increasing the number of image embeddings from 1 to 4 slightly increases the mean acceptance length (τ), this gain is offset by the additional computational cost, resulting in a decreased end-to-end speedup.
>
> We believe it is counterproductive to force a shallow draft model to capture every fine-grained detail. Its purpose is to generate **plausible** token sequences for **the majority of cases** to maximize acceleration. The more difficult cases are efficiently handled by the powerful target model, which is the essence of speculative decoding.
>
>
> [1] W. Han, et al. "ContextNet: Improving Convolutional Neural Networks for Automatic Speech Recognition with Global Context." Proc. Interspeech, 2020.
>
> [2] J. Hu, L. Shen, and G. Sun. "Squeeze-and-Excitation Networks." CVPR, 2018.
>
> [3] J. Devlin, M.W. Chang, K. Lee, and K. Toutanova. "BERT: Pre-training of Deep Bidirectional Transformers for Language Understanding." NAACL, 2019.
>
> [4] A. Dosovitskiy, et al. "An Image is Worth 16x16 Words: Transformers for Image Recognition at Scale." ICLR, 2021.
>
> [5] G. Kim, et al. " Donut: OCR-Free Document Understanding Transformer." ECCV, 2022.

---

> > ### Comment · Reviewer_3b1n · 2025-08-08
> >
> > Thanks the author for the clarifications. For W2, despite the added explanation, the proposed global integration remains conceptually close to prior existing global designs which still undermine the novelty. For W3, please evaluate accuracy/quality evaluation in addition to the speed. The claim that draft errors are rejected, please specify and provide evidence that the outputs(ignored small object) are preserved plus failure cases showing the correction behavior. Overall, the reviewer appreciates the clarifications but intends to maintain the current score.

---

> > > ### Author Response · Authors · 2025-08-08
> > >
> > > Thank you for your follow-up comments and for the opportunity to provide further clarification. We address your remaining concerns below.
> > >
> > > **W2: On the Novelty of Global Feature Integration**
> > >
> > > As we stated previously, the general concept of a *global feature* is well-established, and we do not claim this abstract idea as our core innovation. Our novelty lies in its specific motivation, implementation, and function within the speculative decoding framework for VLMs.
> > >
> > > Furthermore, our contributions extend well beyond this single component. We believe our **primary contribution is the identification and analysis of a fundamental bottleneck** in applying speculative decoding to VLMs: a shallow draft model saturates when processing the long and highly redundant sequences of image patch embeddings. This critical issue was unaddressed by prior work.
> > >
> > > To solve this, we introduce a lightweight **Vision Adapter** to compress redundant visual information and a **Global Feature Injection** mechanism to combat the "lost-in-the-middle" effect that can result from this compression. These architectural innovations are complemented by novel **dataset generation and training strategies** designed to produce high-quality, long-form training data for a robust draft model.
> > >
> > > Our experiments show that ViSpec achieves a speedup of up to **3.2x**, whereas previous methods struggled to surpass **1.5x**. We believe being the first work to achieve meaningful acceleration for VLMs is a strong testament to the novelty and impact of our methodology.

---

> > > ### Author Response · Authors · 2025-08-08
> > >
> > > **W3: On Accuracy Evaluation and Error Correction**
> > >
> > > Our work, ViSpec, is based on the principle of speculative decoding. The token compression and drafting process *ONLY* involves a small, auxiliary **draft model**. The final output is *ALWAYS* verified by the **original, unmodified target model**. The speculative decoding process has been mathematically proven to be **lossless**, meaning it produces an output distribution that is **identical** to that of standard autoregressive decoding from the target model alone [1, 2].
> > >
> > > Because the final output is provably identical, the model's accuracy, quality, and any other performance metrics remain unchanged. Consequently, it is standard practice in the speculative decoding literature to report only acceleration, as other performance metrics are not affected [3-7]. Our paper follows this established convention.
> > >
> > > We understand that we cannot upload images during the discussion period, so we hope the following textual description can clarify the mechanics of the paradigm, specifically addressing your request to "specify and provide evidence that the outputs (ignored small object) are preserved plus failure cases showing the correction behavior".
> > >
> > > Let's consider a simple example. Suppose a user shows the VLM an image containing a small orange cat and asks, *What is this?*
> > >
> > > 1. **Drafting Phase**: The lightweight draft model processes a compressed version of the image. Due to the compression, it might struggle with the fine-grained detail and generate a plausible but incorrect draft: *This is an orange dog*.
> > >
> > > 2. **Verification Phase**: This drafted sequence (*This*, *is*, *an*, *orange*, *dog*) is fed into the large, unmodified target model in a **single forward pass**. Thanks to the causal mask, the target model can verify all tokens in parallel.
> > >
> > > - It processes the **original, uncompressed** image features and recognizes the object is a cat.
> > > - It checks the draft token by token. It confirms that *This*, *is*, *an*, and *orange* are all correct (i.e., the same tokens it would have generated itself).
> > > - When it reaches the fifth token, it sees the draft model's guess is *dog*. The target model's own calculation determines the correct token is *cat*.
> > >
> > > 3. **Correction and Final Output**: The target model **rejects** the draft at the point of the first error (the token *dog*). It accepts the first four correct tokens and then generates the correct fifth token, *cat*. The final, verified output is *This is an orange cat*.
> > >
> > > As this example illustrates, the final output is identical to what the target model would have produced autoregressively. The key difference is efficiency. Without ViSpec, generating these five tokens would require five sequential forward passes in the large model. With ViSpec, the first four tokens were accepted and the fifth was generated all within a **single forward pass** of the target model.
> > >
> > > Crucially, a "draft error" being rejected is *NOT* a failure case; it is the **intended and routine operation** of speculative decoding. The draft model is designed to be fast, not perfect. If the draft model were always correct, we could just use it for inference directly and get a massive speedup for free, which is not feasible. It is precisely this **draft-and-verify** paradigm—where the fast model guesses and the powerful model corrects—that accelerates inference without compromising the quality of the final output. This principle holds true regardless of whether the draft model fails to see a small object or makes any other kind of mistake.
> > >
> > > [1] Y. Leviathan, et al. "Fast Inference from Transformers via Speculative Decoding." ICML Oral, 2023.
> > >
> > > [2] C. Chen, et al. "Accelerating Large Language Model Decoding with Speculative Sampling." DeepMind, 2023.
> > >
> > > [3] X. Liu, et al. “Online Speculative Decoding.” ICML, 2024.
> > >
> > > [4] H. Xia, et al. “Unlocking Efficiency in Large Language Model Inference: A Comprehensive Survey of Speculative Decoding.” ACL, 2024.
> > >
> > > [5] Z. He, et al. “REST: Retrieval-Based Speculative Decoding.” NAACL, 2024.
> > >
> > > [6] H. Xia, et al. “Speculative Decoding: Exploiting Speculative Execution for Accelerating Seq2seq Generation.” EMNLP, 2023.
> > >
> > > [7] Y. Li, et al. “EAGLE-2: Faster Inference of Language Models with Dynamic Draft Trees.” EMNLP, 2024.

---

> > > ### Author Response · Authors · 2025-08-09
> > >
> > > Thanks again for your reply. We would be grateful for the opportunity to address any further concerns you may have.

---

### Note · Authors · 2025-08-14

Dear Reviewers, Area Chairs, and Senior Area Chairs,

We thank you all for your time and valuable feedback. We have carefully considered all the comments and prepared responses accordingly.

Our paper introduces ViSpec, a **speculative decoding** framework designed to accelerate VLMs. Our primary contribution stems from a theoretical analysis of speculative decoding. We demonstrate how a *shallow* draft model’s performance degrades when processing the *long, highly redundant* sequences of image tokens typical in VLMs. This leads to our central argument: **to be effective in a multimodal context, this redundant visual information must be compressed**.

To solve this, ViSpec introduces a synergistic framework. We employ a lightweight **vision adaptor** for token compression and then mitigate the potential *lost-in-the-middle* risk introduced by this compression with a **global visual feature injection** mechanism. This architecture is complemented by a specialized **dataset generation and training pipeline** to create robust draft models. Our approach achieves up to **$3.2 \times$** speedup, far exceeding the **$\approx 1.5 \times$** plateau of prior methods and speaking to its novelty and impact.

We appreciate the opportunity to address the reviewers' comments.
* We thank Reviewer **3Mw6** for their insightful questions, which helped us strengthen the paper.
* We are glad that our experiments on high-resolution benchmarks have addressed the main concerns of Reviewer **mKSj**.
* In response to Reviewer **DJLc**'s query on broader applicability, we demonstrated ViSpec's robustness on high-concurrency and zero-shot video tasks.
* For Reviewer **3b1n**, we clarified that as a **lossless** framework [1, 2], ViSpec's correction mechanism is the proven principle of speculative decoding, which mathematically guarantees an output distribution identical to the target model, making accuracy metrics redundant.

Our work, supported by the theoretical analysis and extensive evidence we have provided, offers a practical advance in efficient VLM inference. We respectfully ask you to consider these points in your final evaluation.

Sincerely,

The Authors of Submission 11762

**References**

[1] Y. Leviathan, et al. "Fast Inference from Transformers via Speculative Decoding." ICML, 2023.

[2] C. Chen, et al. "Accelerating Large Language Model Decoding with Speculative Sampling." DeepMind, 2023.

---

### Decision · Program_Chairs · 2025-09-17

**Decision:**

Accept (poster)

**Comment:**

This paper presents ViSpec, a framework designed to accelerate inference in vision-language models (VLMs) without sacrificing generation quality. The key idea is that draft models struggle with redundant visual tokens, slowing speculative decoding. To address this, ViSpec introduces a lightweight vision adapter that compresses image inputs into compact representations. In addition, it extracts a global feature vector per image and augments text tokens with this feature to preserve multimodal coherence. This dual strategy enables efficient handling of visual information while ensuring alignment between modalities. The authors also propose a new dataset with long-form responses to better support training and evaluation. Experiments demonstrate that ViSpec delivers significant inference speedups across multiple VLM benchmarks.

Reviewers were divided on this submission. They asked many questions about the method and its evaluation.
The detailed answers and additional experiments provided, particularly with video and high-resolution images, as well as the statistics on overheads and speed-up, totally convinced the active reviewers.

The final discussions between the AC and the active reviewers clearly revealed a very positive consensus, and active reviewers  increased or confirmed their positive ratings.

As a result, we are convinced that this submission deserves to be published, and we strongly encourage the authors to incorporate all the suggestions that came out of the rebuttal and discussions into the final version.